# A Qualitative Study on the Care Experience of Emergency Department Nurses during the COVID-19 Pandemic

**DOI:** 10.3390/healthcare9121759

**Published:** 2021-12-19

**Authors:** Hsing-Chi Hsu, Hsin-Ju Chou, Kai-Yu Tseng

**Affiliations:** 1Department of Nursing, Hungkuang University, Taichung 433304, Taiwan; chiqueens@sunrise.hk.edu.tw; 2Department of Emergency, Taichung Veterans General Hospital, Taichung 40705, Taiwan; jn178kimo@yahoo.com.tw; 3Department of Nursing, Central Taiwan University of Science and Technology, Taichung 406053, Taiwan

**Keywords:** emergency department nurses, COVID-19, epidemic prevention, care experience, qualitative study

## Abstract

Background: The rapid spread of the disease has severely impacted healthcare systems around the world. There is a lack of evidence related to the care experience of emergency department nurses. Objective: To understand the care experience and thoughts of emergency department nurses during the COVID-19 epidemic. Methods: Purposive sampling was used to recruit 16 emergency department nurses from a medical center, and a semi-structured interview was used for data collection. The enrollment period was from 28 July 2021 to 30 October 2021. Thematic content analysis was used for data analysis. Results: The care experiences of emergency department nurses during the COVID-19 epidemic can be classified into three themes and six sub-themes. Theme 1: I am the gatekeeper; theme 2: Care and environment challenges: difficulties with equipment and environment, managing patient’s and family members’ emotions, insufficient manpower of care; theme 3: Conflicting emotions: worrying about being infected or transmitting the disease, needs of effective support and empathy, and lack of understanding and discrimination towards the occupation. Conclusion: Emergency department nurses adhered to professional awareness and responsibility during the epidemic and play a critical role in epidemic prevention.

## 1. Introduction

Coronavirus disease-2019 (COVID-19), a disease caused by severe acute respiratory syndrome-coronavirus-2 (SARS-CoV-2), is transmitted from person to person [1] and viral transmission is reduced through full COVID vaccination [2]. The lack of a health data system has restricted the global awareness on the actual mortality rate associated with the COVID-19 pandemic, which has impaired the rapid response planning by countries around the world [3]. More than 200 million people have been diagnosed with the disease globally, and the cumulative death toll is 4 million [3]. The rapid spread of the disease has severely impacted healthcare systems around the world.

The first imported case was confirmed on 21 January 2020, and the Taiwan government established the national Central Epidemic Command Center (CECC). The CECC carried out measures such as border management, limiting medical staff from leaving the country, establishing outdoor triage and consultation zones in hospitals, restricting hospital visits, and formulating hospital standard operating procedures for infectious disease control [4]. In addition, various hospitals have set up COVID-19 rapid screening stations to carry out polymerase chain reaction (PCR) screening for emergency patients, inpatients, individuals accompanying patients, patients with suspected symptoms, such as fever or respiratory tract infection, and other exceptional circumstances [5].

In 2021, nosocomial COVID-19 infection occurred in medical staff. In March, vaccination started, and community infection occurred in May. Consequently, the whole of Taiwan initiated level 3 epidemic alert measures. Members of the public were required to wear masks when they were outside. All schools adopted remote teaching. Dining in was banned, shopping malls and supermarkets implemented crowd control measures, recreation and entertainment venues were closed, large gatherings were banned, and a real-name system was adopted for the public when visiting public places [4]. As a result, emergency PCR screening increased rapidly, and the contingency strategies of medical institutions included: (1) reducing medical operational load; (2) strengthening community surveillance and notification; (3) strengthening employee health surveillance; and (4) suspending international health services to increase manpower and beds. The medical staff in high-risk units, such as emergency departments and specialist wards, were required to undergo nasopharyngeal or deep-throat saliva sampling every 5–7 days. Community screening stations were set up everywhere to decrease infection clusters and to expand the screening volume.

As Taiwan experienced the SARS epidemic, it was once lauded by the world as a model country for epidemic control in 2020, owing to its heightened epidemic control awareness and stringent border controls. However, the raging epidemic overwhelmed Taiwan and the vaccination rate was less than 1% of the population in May 2021 [6]. Due to the rapid increase in confirmed cases, people flocked to emergency rooms to request screening because their activity footprints overlapped with those diagnosed with COVID-19. Until now, over 16,000 people have been diagnosed with COVID-19, and the death toll is 847 [7]. The COVID-19 epidemic has spread rapidly; other countries have reported infection clusters in hospitals, and many nurses have unfortunately been infected and have even died. Globally, healthcare systems have nearly collapsed, due to the high workload of the medical staff and a shortage of trained nursing staff, particular in low- and lower middle-income countries [8,9]. Therefore, maintaining medical capacity and preserving the healthcare system are vital to ensure successful epidemic control [10]. The emergency department is the vanguard of the healthcare system, and is an important department for responding to public health emergencies. During the epidemic, temperature screening, accurate triage, and access control in response to epidemic control policies, disease care, health education, and ethical conflicts [11,12,13] have increased the workload of the emergency department nurses who are under high stress.

Emergency department nurses have had a high risk of infection during the epidemic. Furthermore, they face long nursing hours, are closest to the patients, and must bear the inconvenience and discomfort of wearing protective attire, worrying about spreading the virus, and facing unpredictable work challenges; these conditions show the diverse, complex, and multidisciplinary challenges of emergency department nurses [14]. Attention should be paid to the workload and emotional stress caused by the epidemic [15,16,17,18]. Previous studies have signified that frequent changes in epidemic control policies result in information chaos, and heighten the confusion and burden during nursing operations [16,19]. Medical staff also face discrimination and alienation from other people in their daily lives due to their occupation, which leads to high physical and mental stress [15,17,20,21]. Clear work guidelines, teamwork, epidemic response training, empathy from nursing supervisors, and national support and commitment could help to improve the epidemic control capabilities and care willingness in nurses [12,21,22,23,24].

The epidemic control contributions of nursing staff are widely supported and acknowledged by the society [25]. Although this recognition provides satisfaction and value to nursing work, more attention should be paid to the epidemic’s influence on the nurses [25]. Nurses have played an important supporting role during this epidemic, and may require assistance, as they have been working in a difficult environment. Therefore, enhancing and acknowledging the value of nursing cannot be withdrawn after the epidemic crisis ends [19,24,26]. Researchers are calling for more nursing staff to participate in COVID-19 epidemic control-related studies and discussions to compile and develop the roles of nurses in epidemic control in response to future public health emergencies [15,22,27]. There is a lack of evidence related to the care experience of emergency department nurses. Therefore, the aim of this study is to examine the experience and thoughts of nurses during COVID-19 epidemic control care.

## 2. Materials and Methods

Personal experiences and feelings are unique. Qualitative methods are used to discover the meaning and produce rich descriptions of phenomena in the form of words [28,29,30]. It helps to achieve a deep understanding of personally meaningful interpretation. A qualitative research design was accordingly conducted to explore the participants’ care experiences.

### 2.1. Ethical Considerations

This study was reviewed and approved by the Institutional Review Board (Number: CE21239B). Before the start of the interview, the subjects who met the inclusion criteria of the study were given the “Subject Information and Consent Form”. The investigator informed the subjects about the purpose, procedures, data collection method, participants’ rights, and expected benefits of this study. The emergency department nurses who agreed to participate were enrolled after their consent forms were signed. To maintain the autonomy and confidentiality of the study subjects, all documents were anonymous and coded. The subjects could stop their interviews at any time without stating any reason. The time and venue of the interview were selected by each participant. Due to the infection control policy, all interviews were done outside of the emergency department.

### 2.2. Participants

The subjects of this study were emergency department nurses with previous care experience of patients with COVID-19. Purposive sampling was employed, and a total of 16 subjects agreed to participate in the interview and signed the consent form. The inclusion criteria were: nurses aged ≥ 20 years with Taiwanese nationality and (a) a valid nursing license; (b) working experience in the emergency department for ≥1 year; (c) previously participated in the emergency care of patients confirmed or suspected to have COVID-19; (d) agreed to participate in this study’s thematic interview after receiving the explanation and signing the study participation form; (e) subjects who were able to describe their experience in Chinese. The exclusion criteria were: pregnant emergency department nurses, emergency department nurse supervisors, and emergency department nurse practitioners. The enrollment site was the emergency department of a medical center in Central Taiwan. The mean number of patients treated in the emergency department per day was 162. During the epidemic, the emergency department shouldered the responsibilities of emergency resuscitation, COVID-19 rapid antigen testing, and polymerase chain reaction nucleic acid testing. All patients who required hospitalization or emergency treatment were admitted after their COVID-19 rapid antigen test in the emergency department was confirmed to be negative. Patients who tested positive were quarantined.

### 2.3. Data Collection

In this study, research posters were placarded in meeting rooms and lounge areas in the emergency department to recruit the volunteers. The contact information of the researcher was also provided on the posters. After signing the study consent form, the subjects selected their interview times and venues. The interview sites were comfortable to the subjects and they did not feel disturbed; hence, they freely and fully expressed their experiences and thoughts in a focused manner. Each interview lasted for 60 min. The investigator, who was a qualitative study expert, conducted the interviews. Before the study, the investigator spent 1 month researching emergency department nursing work in order to familiarize herself with standard patterns, the policy changes, and practices. In addition, the investigator formulated the interview guidelines, based on the literature findings and experiences of the clinical nurses. An emergency department nurse and a qualitative study expert jointly discussed and confirmed the interview outline with the investigator. The outline was as follows: (1) Please describe your experiences and thoughts on epidemic control-related work in the emergency department; (2) Please discuss your views on the COVID-19 epidemic; (3) Please discuss your epidemic control care experience in the emergency department during the COVID-19 epidemic. What difficulties did you face during COVID-19 epidemic control care? (4) How do you react when you face difficulties during COVID-19 epidemic control care in the emergency department? (5) Please discuss your experiences with positive thoughts or impression during COVID-19 epidemic control care in the emergency department; (6) Did you receive assistance during COVID-19 epidemic control care in the emergency department? Do you have other needs? (7) Do you have other thoughts that you are willing to share with me?

### 2.4. Data Analysis

In this study, the thematic content analysis proposed by Newell and Burnar [31] was used for data analysis. It provides a clear and simple six-step framework analysis and has been used to explore nurses’ occupational stress [32] and care experience [33,34]. The data collection interviews were recorded, and the key points were documented as notes. After each interview, the recorded content was transcribed within 3 days, and the investigator repeatedly read and annotated the transcripts, immersing themselves in the source data. Open coding was used to categorize the descriptions in the source data that were related to the study theme into meaningful titles or categories before the various coded meaning sets were gathered to generate major themes with common characteristics [31]. To augment the trustworthiness of the study results, members of the study team (including the emergency department nursing expert and qualitative study expert) examined the interview contents to avoid missing important content in the analysis [35]. The data became saturated when all codes met the theme categories. The final analysis results were jointly examined by a qualitative expert and a clinical emergency nursing expert to achieve neutrality and verifiability of the study. All data obtained during the study, including transcripts, voice recordings, and study results, were stored in an independent and locked data cabinet for future validation and to enhance the verifiability of the results.

## 3. Results

There were 16 participants, including 2 men and 14 women, with a mean age of 29.88 years and a mean work experience in the emergency department of 7.25 years. Table 1 shows the general information of the participants. In this study, there were 16 participants, of which two were male nurses (12.5%). The number of interview subjects in other similar studies was 9–21, and the gender ratio of study participants was similar (9.5–25%) [36,37,38,39,40]. There were three themes and six sub-themes for the emergency department nurses’ experiences with epidemic control care during the COVID-19 epidemic (Table 2).

### 3.1. Theme 1: I Am the Gatekeeper

The emergency department is the first station for COVID-19 testing and triaging. As the number of local cases was increasing, outdoor testing stations were set up at the emergency department entrances of many hospitals to prevent confirmed cases from entering enclosed spaces and causing cross-infection. Therefore, all patients were required to undergo detailed TOCC evaluation and nucleic acid testing at outdoor well-ventilated areas. Patients could only enter the indoor spaces of the hospitals to receive treatment after their nucleic acid test results were found to be negative. Suspected or confirmed cases were effectively separated to appropriate sites for testing, quarantine, and care. The participants generally felt that they had a critical mission in infection prevention as they were emergency department nurses preventing the virus from entering the hospitals and causing nosocomial infections, or causing the disease to spread if infected patients were not identified. Therefore, emergency department nurses are gatekeepers in epidemic prevention.

P1 mentioned: “We are the first line of defense in the entire society or hospital. Emergency department nurses are like gatekeepers who must carefully evaluate patients and perform TOCC triaging to identify infected patients and prevent nosocomial infection.” P4 mentioned: “The emergency department is the vanguard, and prevailing policies stipulate that patients must first undergo emergency department screening when entering the hospital. Thus, the emergency department can be regarded as the interception net for blocking COVID-19 from entering the hospital. I am required to enquire about TOCC in detail and filter the patient’s condition layer by layer”.

Due to the rapidly changing epidemic, national epidemic prevention policies and hospital infection prevention policies are continuously updated. Emergency department nurses have demonstrated their professional awareness. The participants mentioned that they had continuously paid attention to epidemic information, understood the infection chain and the footprints of confirmed cases, automatically updated epidemic prevention policies, adopted professional sensitivity when facing all patients and public who sought medical attention at the emergency department, and performed accurate screening and triaging, so that they could keep abreast of changes in epidemic prevention regulations before they went to work.

P6 pointed out: “Every day, I had to pay attention to current affairs to know what stage the epidemic was at the moment, and to update changes to the procedure or hospital spaces…To prevent omissions or errors after I returned from my vacation, I had to maintain alertness to detect anomalies”.

With regards to their first-line role in the emergency department, most participants showed a high degree of self-work and acknowledgment of their roles as emergency department nurses. P5 mentioned: “Everybody is working together to combat the epidemic as we protect the health and safety of the entire hospital, even the whole country together, drenched in sweat together, cared for patients together…I felt that we lived up to the motto of “performing our responsibility without fear and combating the epidemic together” I feel very honored”.

### 3.2. Theme 2: Care and Environment Challenges

There are three sub-themes for this theme, namely, Sub-theme 1: Difficulties with equipment and environment; Sub-theme 2: Managing patient’s and family members’ emotions; Sub-theme 3: Insufficient manpower of care.

#### 3.2.1. Sub-Theme 1: Difficulties in Equipment and Environment

In order to prevent nosocomial infection, an outdoor, well-ventilated testing station was set up outside the emergency department. As the epidemic outbreak occurred in May, which was hot in terms of weather, the testing station that was temporarily set up was a tent or prefabricated building without air conditioning or fans. In addition, emergency department medical staff had to wear personal protective equipment (PPE); thus, all participants mentioned the discomfort caused by the equipment and environment, and the difficulties in performing nursing tasks.

P8 pointed out: “The prefabricated building was very hot and stuffy, and felt like an oven. My discomfort increased with the time spent in the building. Wearing an N95 mask and the entire PPE makes me felt stuffy and uncomfortable”.

P2 mentioned: “The outdoor environment was a simple environment with a lot of difficulties that we must overcome when performing outdoor treatment. For example, I had to wear the PPE, two layers of gloves, and an overall (disposable medical gown). It was already difficult enough to perform intravenous injection in adults, and I had to inject a 3-day old newborn, which was too much for me!”.

P4 stated: “Our negative pressure isolation ward was very small, and we had to wear the entire PPE set. There was chaos when patients required emergency intubation or resuscitation and we had to endure discomfort and inconvenience from wearing the PPE, resulting in great physical and mental burden. This and the monitor alarm cause physical and mental fatigue.

Furthermore, the N95 mask and protective goggles caused breathing difficulties and affected expression and voice transmission during communication. In order to explain to the patient’s family members, more than half of the participants mentioned that they had to raise their voice and this even led to communication barriers.” P6 stated: “I felt like I was quarreling with people all day. I had to raise my voice when I wore the N95 mask, and the patients and family members could not hear me clearly, which made it look like that we were quarreling”.

P12 pointed out: “After we put on the PPE, the patients would feel distant from us and that we were not as friendly. I think that there was a gap between nurses and patients. In most situations, I had to raise my voice to explain to the patient and the patients would misunderstand or feel that I had poor attitude if my tone or volume was not controlled properly”.

#### 3.2.2. Sub-Theme 2: Managing Patient’s and Family Members’ Emotions

During the epidemic, the emergency department had to manage members of the public who came for rapid screening due to epidemic anxiety, on top of critically ill patients, and the work load in the emergency department increased. Due to compliance with epidemic prevention procedures, inconvenience from wearing PPE, and delayed care due to donning and doffing, the emotions of the public became heightened, causing frontline emergency department nurses to face the emotions of patients and their family members.

P4 mentioned: “We had to bear with anxiety and impatience from the public, patients and their family members. The public would ask why we had to conduct tedious TOCC questioning? Why do I have to stay outdoors? Why is there only 1 family member allowed to stay with the patient in the emergency department? Why does the PCR report take so long? Sometimes, they even scold and scream at us.” P10 stated: “environmental difficulties and equipment discomfort caused us to slow down when we performed techniques. In addition, there were many patients in the emergency department and we could only manage emergency cases first while allowing non-emergency cases to wait, which tended to arouse anger in the public. For example, how long do I have to wait? It is very hot outside and how long do I have to stand? Why am I not allowed in the emergency department even though I am not feverish?”.

#### 3.2.3. Sub-Theme 3: Insufficient Manpower of Care

The outdoor quarantine zone is a safe infection prevention policy for emergency department nurses, and the participants all mentioned that this was a safe policy. However, in addition to the general public who were waiting for their PCR test results, there were critically ill patients who sought medical attention, and many medical treatment and care activities were still ongoing. However, the manpower of care was insufficient to meet the demand of the outdoor quarantine zone.

P5 pointed out: “When there was a patient with COVID-related symptoms, we had to leave him/her at the outdoor quarantine zone. Every time there was an announcement of new infection chain, the number of patients in the emergency department would increase but the manpower did not. I could only wait for colleagues from another zone to assist me. I felt very stressful when the patients waiting outside were seriously ill”.

P6 pointed out: “In addition to the general public who awaited PCR screening results in the outdoor quarantine station, there were also many patients who required emergency treatment such as insertion of nasogastric tube and urinary catheterization. One day, I was responsible for as many as 22 patients in the outdoor quarantine zone, of which 2 were in critical condition. One even had intracerebral hemorrhage. Furthermore, there were two bacteremia patients on sepsis bundle. As there was no empty bed in the negative pressure ward in the emergency management, these patients had to undergo emergency procedures at the outdoor quarantine zone, which was too much for me”.

More than half of the emergency department nurses mentioned that, in addition to emergency department nursing work, nurses thought that additional tasks during the epidemic were non-emergency but essential, and that they were even non-nursing tasks.

P15 mentioned: “In epidemic prevention, there are many non-emergency but essential quarantine tasks added to the emergency department, such as asymptomatic screening, health education, explaining the screening procedure, and even soothing the uncontrollable emotions of the public and maintaining order”.

### 3.3. Theme 3: Conflicting Emotions

The sub-themes included: Sub-theme 1: Worrying about being infected or transmitting the disease; Sub-theme 2: Desiring support and empathy; Sub-theme 3: Lack of understanding and discrimination towards the occupation.

#### 3.3.1. Sub-Theme 1: Worrying about Being Infected or Transmitting the Disease

Although emergency department nurses have acknowledged their professional responsibilities and that they should perform their role without fear, being gatekeepers in contact with an emerging disease with no cure and with a high risk of infection, emergency department nurses have an extreme fear of being infected. P12 mentioned: “emergency department nurses are at all high risk and worried that they would be a loophole of epidemic prevention. The subject stayed in the hospital dorm even on vacation, which affected her daily life and socializing”.

P1 pointed out: “After there was a sudden spike in the number of confirmed cases in Taiwan, I have not returned home for more than 1 month. My home is not very far (from the hospital) but I did not return home as my patients are old and I am afraid that I got infected and bring the virus home.” P11 stated: “I contact many patients every day but most are asymptomatic patients. In particular, I worry about getting infected unknowingly when I contact confirmed cases during work. In order to allow medical staff to go to work rest assured, we have to do PCR screening every week where we have to tolerate the discomfort of nasal swab and also worry about being infected. We worry that our professional career will end if we get infected (laughter)”.

#### 3.3.2. Sub-Theme 2: Needs of Effective Support and Empathy

As Taiwan’s epidemic situation was stable for more than a year and there was a sudden spike in the number of confirmed cases, the emergency department’s epidemic prevention cabins were hastily built, and epidemic prevention policies were continuously updated, due to changes in the epidemic situation. Participant P7 mentioned that colleagues in the department worked together in the environment at the start of the epidemic, and there was no supervisor to provide assistance or planning. This made him/her feel isolated and helpless. She/he emphasized that emergency department work requires fast and efficient assistance, and includes understanding of the work of emergency department nurses by the hospital, emergency department, and general public.

Most participants mentioned that support was provided from the emergency department, e.g., P15 mentioned: “In the emergency department, the senior employees are extremely sharp and rapidly established epidemic prevention procedures, arranged education, training, and drills for COVID-19, including infection prevention traffic (infection zone, relatively clean zone, and clean zone), donning and doffing procedure, and negative pressure measures.” At the same time, an information zone was set up using communications software. P10 mentioned: “Our department has its own club and group. When there are new policies released, the duty leader will publish it in the club and group so that our colleagues can quickly obtain information.

Support should also be provided in the form of hospital-directed environment and protective equipment. P13 pointed out: “At the beginning, the outdoor quarantine zone was an open environment, and the weather was hot. After that, tents were built and then prefabricated buildings were built. When we mentioned our problems, the department supervisor or hospital manager would try his/her best to improve and pay particular attention to whether there was sufficient protective equipment. I felt that they provided a lot of support”.

The participants also mentioned societal support and warmth. The participants mentioned that companies and the general public appreciated medical staff for their efforts through actions and words. For example, they provided many medical supplies or food to medical staff. P10 stated: “I have two sons, one in childcare, and one in kindergarten. When policies opened up and allowed childcare, the childcare center informed me that priority was assigned to frontline epidemic prevention staff. Even though the childcare center knew that I was a medical staff, they did not fear nor reject my child. The teachers also mentioned that medical staff are working hard, which made me feel that they were friendly and I was touched by that”.

#### 3.3.3. Sub-Theme 3: Lack of Understanding and Discrimination towards the Occupation

The emergency department is the first line of epidemic prevention, and some emergency department nurse participants mentioned that there was a lack of understanding towards their occupation and felt alienated and discriminated against.

P7: “One of my family members mentioned that ‘Please don’t come home if you are caring for suspected patients’. Although I knew that they did not mean it but I still felt hurt as to why my family members did not understand me.” P5 pointed out: “My younger brother is working in Taipei. When his colleagues found out that I am an emergency department nurse, my brother was asked to work from home by his company. This is a form of discrimination but his colleagues said that it was just a fear rather than discrimination. To me, this felt like discrimination”.

In addition, the government provided *epidemic prevention incentives* as a form of appreciation for the hard work by medical staff. Many participants felt that the emergency department is the gate in the hospital for epidemic prevention, but little attention was paid to the efforts and risks of emergency department nurses during epidemic prevention. P6 mentioned: “I bear the risk of getting infected when I contacted suspected or confirmed cases in the outdoor quarantine zone. I wore the PPE in the emergency department and even throughout the entire shift, which was similar to dedicated ward nurses and could not remove the PPE to rest at the nursing station... (Sigh).” Therefore, the epidemic prevention incentives provided by the country have shortchanged emergency department nurses. P1 pointed out: “We have to constantly face patients and their family members, provide more explanation and communication, and soothe emotions on top of epidemic prevention. I feel that we also play a part in epidemic prevention like dedicated ward nurses but our epidemic prevention incentives are different”.

## 4. Discussion

This study aimed to examine the epidemic control care experience of emergency department nurses during the COVID-19 epidemic. Three themes were obtained from the analysis, namely (1) I am the gatekeeper; (2) Care and environment challenges; and (3) Conflicting emotions. From the 2002 SARS experience to the COVID-19 pandemic, it has been shown that the emergency department is the first line of defense for disaster response, and epidemic control and screening. During the COVID-19 pandemic, emergency measures of medical institutions included emergency vigilance in the form of hospital traffic planning, triaging and control, and adjusted responses of nursing procedures [12].

Nurses with different tasks and work experience would have varying perspectives on epidemic control [24]. This study is the first to examine the COVID-19 epidemic control care experience of emergency department nurses and has unique values. The results suggested that the participants were generally aware of their role in plugging the gaps in epidemic control as emergency department nurses. They constantly paid attention to epidemic information, understood the infection chain and the footprint of a confirmed case, employed professional sensitivity in facing the patients and the public in the emergency department, and performed precise screening and triaging. According to the literature [17,41], nurses who provide COVID-19 care with professional responsibility are aware of the professional performance required by the occupation, as well as their passion for their own occupation. These findings are similar to the results of the present study in which many participants mentioned that they felt honored and proud to be able to participate in COVID-19 epidemic control care. However, a qualitative study on the COVID-19 critical care experience in the USA conducted semi-structured interviews on 11 intensive care unit nurses, where many of the participants did not agree that nurses should be referred to as heroes by the society and the media, and they rejected being called the epidemic control heroes. However, the participants of this Taiwanese study were proud to be involved in epidemic control. A possible reason could be that different people have different interpretations of “heroes” or “honor” [42]. The present study proved that acknowledgment and acceptance by the public toward healthcare workers is one of the important supports for the emergency department nurses.

The fear of becoming infected with the virus is one of the major sources of stress in the emergency department nurses’ experience of epidemic control care. Identical results were obtained in this study and in past studies. Nurses possess professional self-awareness, strictly comply with epidemic control policies during the epidemic, and reduce contact with external family members [10,17,21,43]. At the same time, the nurses understand that epidemic control policies undergo rolling and frequent revisions, and adjust their decisions based on changes in the epidemic [16]. The need for the emergency department nurses to constantly update the rules of epidemic control information has been a work burden to them. This need also tends to cause information confusion and increases epidemic control care stress and distress in the nurses [23,43].

Uncertainty about safety, lack of support, urgent need for care organization and cross-department communication, and demand for personal protective equipment (PPE) are present in other frontline medical staff [36,44,45].

The participants of this study mentioned that sufficient protective equipment was available and that they felt safe during the work of epidemic control [16,24]. Previous papers have mentioned that there was a lack of PPE during the COVID epidemic [46,47]. These studies found that there was insufficient PPE in the initial stages of the COVID-19 pandemic in 2020. Although the participants in this study received sufficient support and protective equipment from the government and the hospital, this may not apply to other grades of hospitals. In addition, the burden caused by protective equipment, such as discomfort and inconvenience of the protective equipment and time-consuming donning and doffing procedures, which were found in this study, were similar to those observed in previous studies [19], and dentists and nurses in this study have identical views as they felt that PPE would impede communication [45]. Furthermore, the present study found that the distance and the amplified volume of speech due to wearing the protective equipment can create difficulties in communication between patients and nurses. This difficulty causes tension in the interaction between the emergency department nurses and patients or family members, and increases the workload of the emergency department nurses. A large number of people needed to undergo screening in the emergency department during the COVID-19 epidemic. Hence, anxiety and the uncomfortable epidemic control environment, and adjustments to the care procedure in response to epidemic control causes the patients and family members to vent their physical and mental discomfort on the emergency department nurses who work in the frontline and provide direct care [10,22]. This study proves that the simultaneous presence of different care rhythms and scenario stress are unique care experiences of emergency department nurses. The nurses need more support to address the delayed care caused by the discomfort and inconvenience of protective equipment and time-consuming donning and doffing procedures. In addition, other staff members were required to assist in setting up the epidemic control environment, explain screening procedures, sooth the emotions of the patients and family members, and assist in patient transfer [20].

The respect and acknowledgment in the society can increase the willingness to work and the self-worth in the nurses [25,41]. This finding is consistent with the results of the present study, whereby the expression of appreciation and support by the society and the public to the medical staff during the epidemic caused the nurses to experience positive support and encouragement. Owing to COVID-19 care, the nurses not only experienced extensive physical and mental stress but also needed to have strong courage and resilience. Even though support is shown by the society and positive energy is transmitted by the media, the nurses still face discrimination and alienation in their living environment due to the occupational risk [15,25]. In the Gorfon et al., study [17,41], the participants expressed that performing epidemic control care caused them to feel isolated, unable to perform other tasks or to visit anyone, and the only place they could go was to work. In contrast to other studies, the emergency department nurses in this study mostly felt that they were an important epidemic control window in the hospital and shouldered a critical role of safeguarding the hospital’s environment. Therefore, they mostly viewed the changes in lifestyle habits based on their gatekeeper role with a positive attitude. In addition, the Taiwanese government has provided monetary incentives to care for the patients confirmed to have COVID-19. However, the participants of this study felt that the incentives did not consider the risk and work characteristics of the emergency department and that their role as emergency department nurses was not valued or acknowledged. Even though they had conflicting views on the value of their occupation, the emergency department nurses still chose to remain in their positions and not run away and hide [17,41].

## 5. Limitations

Owing to time and workforce limitations, the subjects enrolled in this study were limited to one medical center in Taiwan. Because of the differences in traffic planning and epidemic control measures of the emergency department between different grades of hospitals, the results of this study cannot be completely generalized to emergency department nurses from hospitals with other grades. Furthermore, emergency department nursing supervisors and pregnant nurses did not directly participate in the frontline care of patients suspected of or confirmed to have COVID-19, and the work content of specialist nurses differs from that of nurse practitioners. Therefore, the results of this study cannot be generalized to these populations. Future studies can compare physical and mental states between nurses in high-risk departments (such as the emergency department, dedicated wards, and intensive care units) and those not responsible for caring for confirmed cases, and carry out long-term follow-up on the impact of the COVID-19 pandemic on emergency department nurses. Furthermore, COVID-19 pandemic control-related studies require the participation of more emergency department nurses to develop better responsiveness for major infectious disease and other public health emergencies in the future.

## 6. Implication for Practice/Policy

The government should attach importance to the role positioning of emergency department nurses, and emergency department nurses should be included in the first line of defense in public health and receive treatment equal to their colleagues specifically responsible for epidemic control. Nurses and healthcare managers should devote greater attention to the care needs and psychophysiological support of emergency department nurses. During a global crisis such as the COVID-19 pandemic, healthcare policy-makers should limit the number of non-critically ill patients seeking medical attention in the emergency department.

## 7. Conclusions

The emergency department nurses possessed role awareness and professional self-awareness in the experience of COVID-19 epidemic control care, and could strictly follow epidemic control rules and adjust their lifestyle habits. Caring for patients confirmed or suspected to have COVID-19 caused the emergency department nurses to feel highly insecure and to have increased physical and mental stress. Hence, societal understanding and national commitment are needed to improve professional recognition and professional value in the emergency department nurses. When the national epidemic control policies are rapidly changing, the nursing supervisors should possess rapid adaptive capability, observe difficulties in epidemic control care in the emergency department nurses, and assist in the setting up and execution of epidemic control facilities based on the work experience and perceptions of the emergency department nurses.

## Figures and Tables

**Table 1 healthcare-09-01759-t001:** Basic information of participants (*n* = 16).

Participants	Gender	Age	Education Level	ED Working Years
A	F	28	Bachelor	6
B	F	37	Bachelor	14
C	F	27	Bachelor	5
D	F	30	Master	8
E	F	32	Bachelor	9
F	F	29	Bachelor	7
G	F	33	Bachelor	11
H	F	25	Bachelor	2
I	F	32	Bachelor	8.5
J	F	32	Bachelor	10
K	M	30	Bachelor	6
L	M	30	Master	7
M	F	35	Bachelor	13
N	F	27	Bachelor	3.5
O	F	26	Bachelor	4
P	F	25	Bachelor	2

Note: ED: emergency department.

**Table 2 healthcare-09-01759-t002:** Themes and sub-themes.

Theme	Subtheme
I am the gatekeeper	
Care and environment challenges	Difficulties with equipment and environment Managing patient’s and family members’ emotions Insufficient manpower of care
Conflicting emotions	Worrying about being infected or transmitting the disease Needs of effective support and empathy Lacking understanding and discrimination towards the occupation

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
