# Peer review of "A Qualitative Study on the Care Experience of Emergency Department Nurses during the COVID-19 Pandemic"

_healthcare, 2021, doi:10.3390/healthcare9121759_

Round 1

Reviewer 1 Report

Congratulations, good job. Even though, some changes are needed:

  • Please give a reference to line 63 "The epidemic control contributions of nursing staff are widely supported and acknowl-63 edged by the society."
  • I see some disregarded mistakes in the writing when it comes to nurses quoting. If the text is written in direct style, please add quotation marks after colon and remove the word "that", which is often along the results (e..g: in line 165, instead of "P4 mentioned that: The emergency department is the vanguard..." write P4 mentioned: "the emergency department is the vanguard..."). If for any reason you would rather use indirect style, please, review the verbal tenses and remove the colons.
  • Please reconsider renaming sub-theme 1 (line 277) as something like "Worrying about being infected or transmit the disease" since there is one statement from P1 (line 285) that claims to be afraid of "bring the virus home".

Reviewer 2 Report

Thank you for the opportunity to review this manuscript. This paper presents evidence from a qualitative study of semi-structured interviews with 16 Taiwanese emergency department nurses reflecting on their experiences of working through the COVID-19 pandemic. It draws out themes of responsibility, the care environment and its challenges, and the complex emotional experiences of the participants in the conduct of their duties, and makes some concluding recommendations based on the findings of the study.

This study has high levels of timeliness/significance and potential reader interest, and at its core is a very good paper with merit for publication.

However, there is much in the paper that needs more work, either to provide more context or clarify existing information. The background scope of the paper needs a more solid international grounding to be suitable for publication in an international journal. Some more work needs to be done in the Introduction in order to better situate this paper in the global context before focusing on the Taiwanese setting, and better contextualise the study’s Taiwan focus. The methods need some slight reinforcement to connect them to wider literature and justify them, and clarification on their application in the study. The presentation of the findings needs to better pinpoint the participants’ own voices within its narrative. The discussion needs to consider its findings in a broader body of studies on HCW experiences of COVID-19 and ground all conclusions effectively. More could be done in terms of recommendations both for policy/practice and for scholarship.

Please see attached for detailed specific suggestions.

Author Response

Many thanks for your comment. We want to thank you for taking time from your busy schedule to comment on our manuscript.

Reviewer 3 Report

The article presents the results of a qualitative investigation whose approach and findings are relevant on the context of pandemic health care.

Author Response

Dear Reviewer,

Many thanks for your comment. We want to thank you for taking time from your busy schedule to comment on our manuscript.

Round 2

Reviewer 2 Report

Well done to the authors for taking on the revisions suggested by the reviewers thoroughly. I recommend this piece now be published, with congratulations on an important and insightful contribution to the COVID-19 literature.